

# Predatory functional response and fitness parameters of *Orius strigicollis* Poppius when fed *Bemisia tabaci* and *Trialeurodes vaporariorum* as determined by age-stage, two-sex life table

Shakeel Ur Rehman[1], Xingmiao Zhou[1], Shahzaib Ali[1], Muhammad Asim Rasheed[1], Yasir Islam[1], Muhammad Hafeez[2], Muhammad Aamir Sohail[3] and Haris Khurram[4]

[1] Hubei Insect Resources Utilization and Sustainable Pest Management Key Laboratory, College of Plant Science and Technology, Huazhong Agricultural University, Wuhan, China
[2] State Key Laboratory Breeding Base for Zhejiang Sustainable Pest and Disease Control, Institute of Plant Protection and Microbiology, Zhejiang Academy of Agricultural Sciences, Hangzhou, China
[3] The Key Lab of Plant Pathology of Hubei Province, Huazhong Agricultural University, Wuhan, Hubei, China
[4] Department of Sciences and Humanities, National University of Computer and Emerging Sciences, Chiniot-Faisalabad Campus, Chiniot, Pakistan

Corresponding author
Xingmiao Zhou,
xmzhou@mail.hzau.edu.cn

## ABSTRACT

**Background**. The polyphagous predatory bug *O. strigicollis* is an active predator used to control thrips and aphids. The whitefly species *Bemisia tabaci* and *Trialeurodes vaporariorum* are voracious pests of different economic agricultural crops and vegetables.
**Method**. In this study, the Holling disc equation and the age-stage, two-sex life table technique were used to investigate the functional response and biological traits of third instar nymphs and adult female *O. strigicollis* when presented third instar nymphs of both whitefly species as prey.
**Results**. The results showed a type II functional response for each life stage of *O. strigicollis* when fed each whitefly species. The calculated prey handling time for different *O. strigicollis* life stages were shorter when fed *T. vaporariorum* than when fed *B. tabaci* nymphs. In contrast, the nymphal development of *O. strigicollis* was significantly shorter when fed *B. tabaci* than *T. vaporariorum* nymphs. Additionally, the total pre-oviposition period of adult females was statistically shorter when fed *B. tabaci* nymphs than *T. vaporariorum* nymphs. Furthermore, the survival rates and total fecundity of *O. strigicollis* were higher when fed *B. tabaci* than *T. vaporariorum*. There were no significant differences in any population parameters of *O. strigicollis* when fed either whitefly species. These results show that *O. strigicollis* could survive and maintain its populations on both species of whitefly and could therefore serve as a biological control agent in integrated pest management (IPM).

## INTRODUCTION

Invasive insect pests can significantly disturb native insect communities and cause considerable damage to agriculture and forests (*Pimentel et al., 2000*). Among these pests, whiteflies (Hemiptera: Aleyrodidae) are the most damaging insect pests to agricultural crops globally, including China where more than 1450 species are known (*Martin, Mifsud & Rapisarda, 2000*; *Anderson et al., 2004*; *Lapidot et al., 2014*). Included in these species are the silver-leaf whitefly (*Bemisia tabaci*) and greenhouse whitefly (*Trialeurodes vaporariorum*), which are generally considered responsible for major economic losses. However, *B. tabaci* is thought to be a complex of morphologically indistinguishable sibling species, referred to as different biotypes (*Wraight, Lopes & Faria, 2017*). They are considered to be one of the most significant plant pests and colonize over 600 host plants, causing significant damage (*Polston, Barro & Boykin, 2014*). This species of whitefly is distributed globally (*De Barro, 1995*; *De Barro et al., 2000*; *De Barro, Trueman & Frohlich, 2005*); and causes significant damage to crop yield and quality by feeding on plant phloem and secreting honeydew that stimulates the rapid growth of molds (*Colvin et al., 2006*; *Prijović et al., 2013*). However, the most significant problem associated with outbreaks of *B. tabaci* is the transmission of plant viruses (*Navas-Castillo, Fiallo-Olivé & Sánchez-Campos, 2011*). *Bemisia tabaci* has been reported as the vector for the transmission of over 300 viral species in major economically important agricultural and vegetables crops (*Gilbertson et al., 2015*). Similarly, *Trialeurodes vaporariorum*, commonly known as greenhouse whitefly, is also considered an important pest of vegetable and agricultural crops but transmits fewer viruses than does *B. tabaci* (*Wisler et al., 1998*; *Jones, 2003*; *Brown, 2007*; *Navas-Castillo, Fiallo-Olivé & Sánchez-Campos, 2011*; *López et al., 2012*). However, because of its short life cycle, this species has been considered a more prevalent insect pest in greenhouse (*Simmonds et al., 2002*). *Trialeurodes vaporariorum* can also adapt to cold climates better than *B. tabaci* does and is common at high elevations (*Barboza et al., 2019*). Because of its resistance to insecticides such as neonicotinoids, *T. vaporariorum* has received much research attention (*Gorman et al., 2007*; *Karatolos et al., 2011*; *Pym et al., 2019*).

Extensive use of pesticides not only causes environmental contamination and ozone layer depletion, but also creates serious health problems in mammals and creates toxic conditions for beneficial insect species (*Shaaya et al., 1997*; *Yoza et al., 2005*). Hence, to reduce insecticide use, biological control methods such as use of natural predators, resistant varieties, and plants extracts are important in controlling pests in modern integrated pest management (IPM) programs (*Kageyama et al., 2010*; *Yazdani & Zarabi, 2011*; *Yang, Zhu & Lei, 2012*; *Asare-Bediako, Addo-Quaye & Bi-Kusi, 2014*).

The genus *Orius* (Heteroptera: Anthocoridae) is the largest group of flower bugs, containing around eighty species globally. They are polyphagous predators of small and soft-bodied insects considered pests in agriculture and forestry, including spider mites, aphids, thrips, and whiteflies in protected and open-field crops within its native range of Asia (*Herring, 1966*; *Hernández, 1999*; *Postle, Steiner & Goodwin, 2001*; *Carpintero, 2002*; *Arnó, Roig & Riudavets, 2008*; *Yamada, Yasunaga & Artchawakom, 2016*; *Zhao et al., 2017*). The artificial introduction of *O. sauteri* as a biological control agent provides potential control

against small insect pests on pepper and eggplant, especially under greenhouse conditions (*Jiang et al., 2011*; *Yin et al., 2013*). *Orius laevigatus* is an effective biological control species in Europe and is widely used in augmentative release programs (*Van Lenteren & Bueno, 2003*). Studies have been conducted into predation by *O. albidipennis, O. insidiosus, O. majusculus*, and *O. niger* on different prey species (*Fritsche & Tamo, 2000*; *Tommasini, Lenteren & Burgio, 2004*; *Rutledge & O'Neil, 2005*). Similarly, *O. majusculus* and *O. laevigatus* were reported as potential natural enemies of *B. tabaci* eggs, nymphs, and adults (*Arnó, Roig & Riudavets, 2008*).

The predatory flower bug *Orius strigicollis* Poppius (Heteroptera: Anthocoridae) which was previously known as *Orius similis* Zheng (Heteroptera: Anthocoridae) (junior synonym of *O. strigicollis*) (*Yasunaga, 1997*; *Jung, Yamada & Lee, 2013*), also called "minute predatory flower bug", is a common and effective natural enemy present in cultivated fields of China and is used as a biological control agent against many small pests of economically important agricultural crops (*Zhang et al., 2012*). Both pre-adult and adult stages prey on lepidopteran insects including eggs or newly hatched larvae of *Pectinophora gossypiella*, *Anomis flava*, and *Helicoverpa armigera* as well as *Frankliniella formosae, Aphis gossypii*, and *Tetranychus cinnabarinus*. Large populations of *O. strigicollis* in cotton fields are useful as biological control agents (*Zhou & Lei, 2002*). This Anthocorid species has many features that make it a good biological control agent, such as high searching efficiency, the ability to increase population levels with outbreaks coinciding with prey density, and an aptitude to aggregate in regions of high prey populations (*Hodgson & Aveling, 1988*). Mass rearing of *O. strigicollis* and the subsequent augmentative release into crop fields leads to the control of many insect pests, decreasing their populations and hence reducing the use of pesticides (*Tommasini, Lenteren & Burgio, 2004*; *Bonte & De Clercq, 2011*). However, it is important to estimate the effectiveness of a predator before using it in an integrated pest program (*Fathipour et al., 2006*).

The potential of a predator to control a pest depends upon its functional response to different populations of prey (*Butt & Xaaceph, 2015*). Therefore, the efficiency of a predator can be assessed by its functional response (i.e., changes in attack rate in response to variations of prey populations × number of prey consumed per unit time in relation to prey density (*Riechert & Harp, 1987*). Four types of functional response have been defined based on the predation rate of a predator as a function of prey density: type I (a linear increase), type II (an increase with a slowdown at high prey densities), type III (a sigmoidal increase) and type IV (a dome shape in prey consumption increase (*Holling, 1961*; *Pervez, 2005*; *Sakaki & Sahragard, 2011*). Similarly, the biological traits of a predator, influenced by changes in prey species, greatly affect its predation activity. Thus, this study aims to investigate the interaction of the predator *O. strigicollis* with the prey species *B. tabaci* and *T. vaporariorum* under controlled conditions. The main aim is to evaluate the functional response parameters, fitness parameters, biological traits, and population parameters of *O. strigicollis* when fed *B. tabaci* and *T. vaporariorum* nymphs separately.

## MATERIAL AND METHODS

### Insect rearing

Adults of *O. strigicollis* were captured from vegetable and cotton fields of the Huazhong Agricultural University (Wuhan, China) and mass reared in an insectarium following the method described by *Zhou et al. (2006)* with slight modifications. The rearing arenas consisted of transparent boxes (23.5 × 22.0 × 5.5 cm) with ventilation in the lid. Nymphs and adults of *O. strigicollis* were supplied black aphids (*Aphis fabae*). Small stems of *Vitex negundo* (3–4) wrapped with wet cotton over the end were provided as oviposition substrate. The environmental conditions of 28 ± 1 °C temperature, 70 ± 5% R.H and a photoperiod of 16 L: 8 D h at a light intensity of 1400–1725 lux.

Adult *B. tabaci* were collected from vegetables grown in greenhouses and from other crops from open fields located in the campus of the Huazhong Agricultural University. They were then moved to insectaria and released on potted cotton (*Gossypium hirsutum*) plants (10 cm) to develop the stock culture for the experiments (*Khan & Wan, 2015*; *Tomar, Sharma & Malik, 2017*). The stock culture of *T. vaporariorum* was maintained from a few adults received from the Southwest University (Chongqing, China). Large screen cages (65× 65× 65 cm) were used as arenas for both whitefly species. The obtained *T . vaporariorum* individuals were released on tobacco (*Nicotiana tabacum*) plants (10 cm) for mass rearing (*Haiyan et al., 2017*; *Wei et al., 2018*). The following environmental conditions were maintained inside the insectaria: temperature 26 ± 1 °C, RH 65 ± 5%, and a photoperiod of 16 L: 8 D h at a light intensity of 1400–1725 lux.

### Functional Response of *O. strigicollis*

The third instar nymphs and three-day old adult females of *O. strigicollis* were collected from the insectaria and fed third instar nymphs of *B. tabaci* and *T. vaporariorum* for 48 h separately and starved for 24 h. The predatory bugs were then individually transferred to small petri dishes (9 cm in diameter and 2 cm in depth) and supplied separately with third instar nymphs of *B. tabaci* and *T. vaporariorum* with different densities (4, 6, 8, 10, 12, and 14) per predator. After 24 h, predators were removed, and the prey consumed by both life stages of *O. strigicollis* counted under stereomicroscope (*Liu et al., 2018*). Bugs were used once only. All the dead/empty nymphs of both whitefly species were assumed killed by the predator as preliminary study indicated 100% survival of bugs in the absence of whitefly nymphs. Thirty replications of the experiments involving the third instar nymphs and adult females of *O. strigicollis* were made for each treatment/density with both prey species separately.

### Life table study
#### Nymphal development

Approximately sixty freshly laid healthy eggs of *O. strigicollis* were isolated from the insectaria and incubated until hatched. All collected eggs of *O. strigicollis* were equally distributed to feed on *B. tabaci* and *T. vaporariorum* third instar nymphs separately. After hatching, *O. strigicollis* neonates (≤24 h) were isolated in small Petri dishes (diameter: nine cm; depth: two cm) firmed with filter paper. From the results of functional responses, we

 

supplied fifteen third instar nymphs of *B. tabaci* and *T. vaporariorum* separately to each individual of the predatory bug as food. Dead/empty nymphs of whitefly were replaced every day. A stem of *V. negundo* was placed in each Petri dish to provide shelter and moisture to the predatory bugs. The end of each stem was wrapped with wet cotton to keep them moist. Thirty nymphs were used in the experiment with three biological replicates for each prey species. Developmental time for each nymphal instar was measured. Individuals that died before reaching the adult stage were also recorded. Sex was confirmed as soon as the adults emerged.

### Adult longevity and fecundity

Newly emerged male and female *O. strigicollis* adults were paired for mating. Females that mated for more than 1.5 min were considered to have been mated (*Butler & O'Neil, 2007*). Each mated female was placed separately in a new cylindrical translucent vial (2.5 × 14 cm diameter and length respectively) enclosed with an adequate mesh nylon screen. A small section of *Vitex negundo* stem was offered to each *O. strigicollis* female as an oviposition substrate. Each stem was wrapped with moist cotton at the end to provide moisture to the stem as well as the bugs using the method of *Zhou et al. (2006)*. *B.tabaci* and *T. vaporariorum* nymphs ($n = 15$) were supplied into each vial as a source of food for female *O. strigicollis*. Stems of *Vitex negundo* were examined under a stereomicroscope (15×) to confirm egg laying. The stem was changed every day after the female laid the first egg. The total number of eggs laid by each female was counted under a stereomicroscope (15×). All predatory bugs were observed until they died. Development period, survival rate, pre-oviposition and oviposition period, fecundity, and longevity of female and male adults of *O. strigicollis* were recorded.

## Data analysis
### Functional response

The shape of the curve was find using the polynomial logistic regression following the method describe by *Pervez (2005)*, *Butt & Xaaceph (2015)* and *Varshney, Budhlakoti & Ballal (2018)*. The polynomial function described the relationship between the proportion of prey consumed ($N_a$) in relation to the density of prey offered ($N_0$) (*Holling, 1959*). Hence, a cubic model was applied in a logistic regression analysis (*Juliano, 2001*).

$$\frac{N_a}{N_0} = \frac{\exp\left(P_o + P_1 N_o + P_2 N_o^2 + P_3 N_o^3\right)}{1 + \exp\left(P_o + P_1 N_o + P_2 N_o^2 + P_3 N_o^3\right)}$$

In this equation, $P_0$, $P_1$, $P_2$, and $P_3$ represent the intercept, linear, quadric, and cubic coefficients, respectively. The negative and positive values of linear coefficient ($P_1$) define type II and III functional responses, respectively. The cubic expression will often provide a good fit to a type II and III responses and will provide a good starting point for fitting a logistic regression (*Trexler, McCulloch & Travis, 1988*; *Trexler & Travis, 1993*; Juliano, 2001). Modifying the Holling disc equation through reciprocal linear transformation, handling time ($T_h$) and the attack rate (*a*) were calculated (*Livdahl & Stiven, 1983*). This method is famous because of its simplicity and accuracy (*Veeravel & Baskaran, 1997*; *Pervez*

*& Omkar, 2003; Omkar & Pervez, 2004*). The modified Holling equation, after reciprocal linear transformation is as follows:

$$1 / N_a = 1 / \alpha T N_0 + T_h / T + \epsilon$$

where $N_a$ is the number of prey killed by predators during time (T: 24 h). $N_0$ is the density of prey, $\alpha$ is the attack rate and $T_h$ is the predator handling time for one prey item. Hence, the variables used in the regression were $y = ax + b + \epsilon$. The inverse mathematical function for curve estimation was used to estimate the values of $\alpha$ and $T_h$ (*Ali, Naif & Huang, 2011*). For each prey density we calculated total handling time ($Th_{total} = T_h \times N_a$), search time ($Ts = T - T_{htotal}$), attack rate ($\alpha = N_a / (N_0 \times Ts)$), and search efficiency ($E = N_a/N_0$) (*Hassell, 2000; Rocha & Redaelli, 2004*). All statistical analyses were performed in MINITAB 19.

### Life table analysis

The data obtained from experiments were analyzes based on the age-stage, two-sex life table theory using Twosex-MSChart, a computer program (*Chi, 1988; Chi & Liu, 1985; Huang & Chi, 2011; Qi Chen et al., 2017; Farooq et al., 2018*). The different biological traits and demographic parameters such as eggs, nymphs and adult development, oviposition, pre-oviposition, fecundity and $r$, $\lambda$, $R_0$, GRR and $T$ were calculated by following the methodology of *Chi (2015)*.

In the age-stage, two-sex life table, the age-specific survival rate ($l_x$) and fecundity ($m_x$) were calculated using the Eqs. (1) and (2) respectively.

$$l_x = \sum_{j=1}^{k} s_{xj} \tag{1}$$

$$m_x = \frac{\sum_{j=1}^{k} s_{xj} f_{xj}}{\sum_{j=1}^{k} s_{xj}} \tag{2}$$

In this equation, $S_{xj}$ is the age-stage specific survival rate of O. strigicollis ($x$ = age which is in days and $j$ = stage) which describe the probability of survival of neonate to age $x$ and stage $j$, meanwhile, $f_{xj}$ describe age-stage specific fecundity of adult female (*Chi & Liu, 1985*). The Euler–Lotka equation was used to estimate the intrinsic rate of increase ($r$) following the method of iterative division with the age index from 0 using Eq. (3) (*Goodman, 1982*).

$$\sum_{x=0}^{\infty} e^{-r(x+1)} l_x m_x = 1 \tag{3}$$

The values of ($R_0$) which described the ability of the female to produce a mean number of offspring in his lifetime, was calculated as

$$\sum_{x=0}^{\infty} l_x m_x = R_0 \tag{4}$$

The relation between female and $R_0$ can describe as

$$R_0 = F \frac{N_f}{N} \tag{5}$$

Here $N$ and $N_f$ represent the total number of *O. strigicollis* used in the experiment and the number of female adults respectively (*Chi, 1988*). The gross reproduction rate *GRR* and rate of increase ($\lambda$) were estimated as

$$GRR = \sum_{x=0}^{\infty} m_x \qquad (6)$$

$$\lambda = er \qquad (7)$$

The mean generation time ($T$) that a population required to rise to $R_0$-fold of its size i.e., $e^{rT} = R_0$ or $\lambda^T = R_0$ at the stable age-stage distribution was intended using the equation

$$T = \frac{\ln R_0}{r} \qquad (8)$$

The bootstrap test method (100,000 bootstrap) was used to estimate the standard errors of developmental time of each stage, fecundity, adult longevity and population parameters (*Akca et al., 2015*; *Akköprü et al., 2015*; *Tuan et al., 2016*). To compare the values, a pair bootstrap test based on the confidence of the interval of difference was used (Chi, 2015; *Pru et al., 2015*). To obtain the curves of survival rate, fecundity, life expectancy and reproductive vales, Sigma Plot 12.0 was used.

## RESULTS

### Functional response of *O. strigicollis*

The results of the logistic regression analysis for third instar nymphs of *O. strigicollis* were highly significant ($P < 0.05$) suggesting a type II functional response as the linear coefficient (P1) was negative against nymphs of both *B. tabaci* and *T. vaporariorum* (Table 1). Similarly, the adult females also showed a type II functional response against both prey species, although the parameters were not significant.

The functional response curves of different life stages of *O. strigicollis* to third instar nymphs of *B. tabaci* and *T. vaporariorum* at different densities are show in Fig. 1. The number of nymphs of both prey species consumed by third instar nymphs and adult females of *O. strigicollis* increased with increases in the prey density from 4 to 8 nymphs per predator, but plateaued with no significant increase in prey consumption with densities of more than 8 nymphs per predator. When only four nymphs of *B. tabaci* and *T. vaporariorum* were provided, the third instar nymphs of *O. strigicollis* consumed a mean of 3.8 ±0.097 and 3.3 ±0.128 nymphs per predator per day, respectively, indicating that the predator is more efficient at finding whitefly nymphs at low prey densities. When provided with third instar nymphs of *B. tabaci* and *T. vaporariorum*, the maximum and minimum prey consumption levels of third instar nymphs of *O. strigicollis* were (95% and 46.4%) and (82.5% and 44%), respectively. However, there were no significant differences found in maximum prey consumption by adult female *O. strigicollis* (92% and 92.5%) when preying on nymphs of *B. tabaci* and *T. vaporariorum*, respectively. The minimum percentage of *B. tabaci* and *T. vaporariorum* nymphs killed by adult female *O. strigicollis* was 48% and 51%, respectively.

**Table 1  Results of logistic regression for different life stages of *O. strigicollis* against different prey densities.** The negative values of linear coefficient for both life stages of *O. strigicollis* fed on different densities of two whitefly species described type II functional response curve. *P*-value $\geq$ 0.05 is considered as significant. Goodness of fit test with *p*-value $\leq$ 0.05 is best fitted.

| Prey species | Parameters | Estimates | | S.E. | | Z-value | | P-value | |
|---|---|---|---|---|---|---|---|---|---|
| | | 3rd instar | Adult | 3rd instar | Adult | 3rd instar | Adult | 3rd instar | Adult |
| *B. tabaci* | Intercept | 8.1005 | 6.4071 | 1.937 | 1.916 | 4.18 | 3.34 | 0.000 | 0.001 |
| | Linear | −2.024 | −1.3928 | 0.664 | 0.661 | −3.05 | −2.11 | 0.002 | 0.035 |
| | Quadratic | 0.1832 | 0.1176 | 0.072 | 0.072 | 2.55 | 1.64 | 0.011 | 0.102 |
| | Cubic | −0.0058 | −0.0037 | 0.002 | 0.002 | −2.33 | −1.48 | 0.02 | 0.139 |
| | Goodness-of-Fit | | | | | | | | |
| | Deviance | 3.0612 | 1.7498 | | | | | 0.216 | 0.417 |
| | Hosmer-Lemeshow | 3.0418 | 1.7443 | | | | | 0.551 | 0.783 |
| *T. vaporariorum* | Intercept | 3.5934 | 4.3138 | 1.524 | 1.918 | 2.36 | 2.25 | 0.018 | 0.025 |
| | Linear | −0.7081 | −0.5958 | 0.548 | 0.668 | −1.29 | −0.89 | 0.197 | 0.373 |
| | Quadratic | 0.0532 | 0.0341 | 0.061 | 0.073 | 0.87 | 0.47 | 0.387 | 0.64 |
| | Cubic | −0.0016 | −0.001 | 0.002 | 0.003 | −0.74 | −0.38 | 0.461 | 0.704 |
| | Goodness-of-Fit | | | | | | | | |
| | Deviance | 0.3479 | 0.5546 | | | | | 0.84 | 0.758 |
| | Hosmer-Lemeshow | 0.3487 | 0.5593 | | | | | 0.986 | 0.967 |

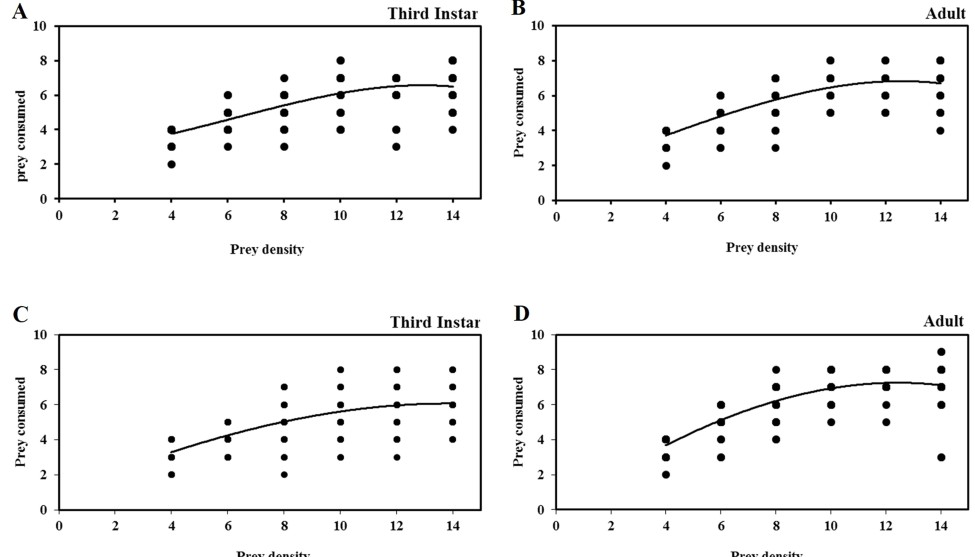

**Figure 1  Functional response of life stages of *O. strigicollis* to different densities of whitefly species.** Each data point represents the observed number of whitefly nymphs. The functional response curves of different life stages of *O. strigicollis* to third instar nymphs of *B. tabaci* (A & B) and *T. vaporariorum* (C & D) at different densities shown type II functional response. The number of nymphs of both prey species consumed by *O. strigicollis* increased with increases in the prey density from 4 to 8 nymphs per predator, but plateaued with no significant increase in prey consumption with densities of more than 8 nymphs per predator.

**Table 2** **Parameter estimates from the Hollings Disc equation for a Type II functional response.** No significant difference in $T_h$ and $T/T_h$ were estimated for third instar nymphs of *O. strigicollis*. However, it was shortest and maximum when adult female fed on *T. vaporariorum* respectively. No significant difference recorded in coefficient of attack rate ($\alpha$) for both prey species.

| Parameters | 3rd instar | | Adult | |
|---|---|---|---|---|
| | *B. tabaci* | *T. vaporariorum* | *B. tabaci* | *T. vaporariorum* |
| Handling Time ($T_h$) | 2.453 | 2.443 | 2.158 | 1.752 |
| Attack rate ($\alpha$) | 0.063 | 0.052 | 0.059 | 0.054 |
| Max. predation rate ($T/T_h$) | 9.78 | 9.82 | 11.12 | 13.70 |

## Functional response parameters

The parameters of functional response (handling time ($T_h$), attack rate ($a$), and maximum predation rate) of different stages of *O. strigicollis* against whitefly species are listed in Table 2. There were no differences estimated in the $T_h$ of third instar nymphs of *O. strigicollis* against both whitefly species. However, adult females undertook shorter handling times (1.75 h) when pursuing, subduing, and consuming *T. vaporariorum* third instar nymphs when compared with the handling times for *B. tabaci* (2.45 h). In contrast, the coefficient of attack rate was higher, ranging from 0.05 to 0.06, when third instar nymphs and adult females of *O. strigicollis* were fed nymphs of *B. tabaci* compared to *T. vaporariorum* (Table 3). However, there were no significant difference noted for different life stages of *O. strigicollis* in attack rate. The maximum predation rate ($T/T_h$) of third instar nymphs of *O. strigicollis* per individual was higher (9.82 d$^{-1}$) for *T. vaporariorum* nymphs compared to *B. tabaci* (9.78 d$^{-1}$). Similarly, the maximum predation rate of adult females was (13.70 d$^{-1}$) and (11.12 d$^{-1}$) for *T. vaporariorum* and *B. tabaci* nymphs, respectively. The functional response parameters of different life stages of *O. strigicollis* to different densities of whitefly species are given in Table 3. The $T_h$ for third instar nymphs and adult females of *O. strigicollis* increased with increasing prey densities. However, the searching time and searching efficiency show an inverse relationship with both prey densities. The attack rates of both life stages of *O. strigicollis* were similar with no significant differences at different densities of both prey species.

## Growth, development, and longevity On *O. strigicollis*

Developmental characteristics of *O. strigicollis* when fed on whitefly species (*B. tabaci* and *T. vaporariorum*) are listed in Table 4. The development time for nymphal instar of N1–N4 of *O. strigicollis* took statistically longer when fed *T. vaporariorum* than when fed *B. tabaci*.

However, there were no significant differences in development of fifth instar *O. strigicollis* nymphs against both whitefly species. In addition, higher mortality rates were observed in the fifth instar when fed *T. vaporariorum* nymphs than when fed *B. tabaci*. The results also showed that longevity of both male and female *O. strigicollis* was statistically similar when *B. tabaci* and *T. vaporariorum* were provided as prey (Table 4). It was also observed that more individuals of the predatory bug survived and successfully reached the adult stage when fed *B. tabaci* nymphs (86.66%) than when fed *T. vaporariorum* nymphs (56.70%).

**Table 3** Functional response parameters of different life stages of *O. strigicollis* to different densities of *B. tabaci* and *T. vaporariorum*. The $T_h$ for third instar nymphs and adult females of *O. strigicollis* increased with increasing prey densities. However, the searching time and searching efficiency show an inverse relationship with both prey densities. The attack rates of both life stages of *O. strigicollis* were similar with no significant differences at different densities of both prey species.

| Prey species | Prey density | Total handling time ($T_h$) | | Total search time (Ts) | | Attack rate ($a$) | | Search efficiency (E) | |
|---|---|---|---|---|---|---|---|---|---|
| | | **3rd Instar** | **Adult** | **3rd Instar** | **Adult** | **3rd Instar** | **Adult** | **3rd Instar** | **Adult** |
| *B. tabaci* | 4 | 9.2 | 8.1 | 14.8 | 15.9 | 0.064 | 0.075 | 0.94 | 0.93 |
| | 6 | 11.3 | 10.4 | 12.7 | 13.6 | 0.060 | 0.070 | 0.77 | 0.80 |
| | 8 | 12.1 | 12.3 | 11.1 | 11.7 | 0.059 | 0.071 | 0.66 | 0.71 |
| | 10 | 15.5 | 14.3 | 8.5 | 9.7 | 0.075 | 0.069 | 0.63 | 0.66 |
| | 12 | 15.6 | 14.4 | 8.4 | 9.6 | 0.063 | 0.058 | 0.53 | 0.56 |
| | 14 | 16.0 | 14.5 | 8.0 | 9.5 | 0.059 | 0.051 | 0.47 | 0.48 |
| *T. vaporariorum* | 4 | 8.1 | 6.5 | 15.9 | 17.5 | 0.053 | 0.053 | 0.83 | 0.93 |
| | 6 | 10.3 | 8.8 | 13.7 | 15.2 | 0.055 | 0.055 | 0.70 | 0.84 |
| | 8 | 12.5 | 11.0 | 11.5 | 13.0 | 0.060 | 0.060 | 0.64 | 0.78 |
| | 10 | 13.6 | 12.2 | 10.4 | 11.8 | 0.059 | 0.059 | 0.56 | 0.70 |
| | 12 | 14.6 | 12.6 | 9.4 | 11.4 | 0.052 | 0.052 | 0.50 | 0.60 |
| | 14 | 14.8 | 12.5 | 9.2 | 11.5 | 0.044 | 0.044 | 0.43 | 0.51 |

**Table 4** Effects of whitefly species as prey on biological traits of *O. strigicollis*. The developmental time for N1, N2 and N4 was significantly higher when fed on *T. vaporariorum*. However, no significant difference recorded in adult longevity. SEs were estimated using 100,000 bootstrap. Means marked with multiple letters in the same row symbolize the significant difference using a pair bootstrap test. $P < 0.05$.

| Life stages | *B. tabaci* Mean ± S.E. | *T. vaporariorum* Mean ± S.E. |
|---|---|---|
| Egg duration (d) | 3.00 ± 0.00 a | 3.00 ± 0.00 a |
| N1 duration(d) | 2.10 ± 0.06 b | 2.30 ± 0.09 a |
| N2 duration (d) | 3.24 ± 0.08 b | 3.80 ± 0.09 a |
| N3 duration (d) | 3.00 ± 0.00 a | 2.96 ± 0.10 b |
| N4 duration (d) | 2.15 ± 0.46 b | 2.42 ± 0.01 a |
| N5 duration (d) | 3.65 ± 0.49 a | 3.71 ± 0.14 a |
| Male adult longevity (d) | 11.67 ± 1.45 a | 11.33 ± 1.52 a |
| Female adult longevity (d) | 15.47 ± 1.30 a | 13.82 ± 1.86 a |

## Fecundity and oviposition of female adults

No significant effects of the presence of the two whitefly species were observed on adult pre-ovipositional period (APOP) of *O. strigicollis* (2.35 days for *B. tabaci* and 2.4 days for *T. vaporariorum*; Table 5). However, the total pre-ovipositional period (TPOP) of *O. strigicollis* was significantly longer when offered *T. vaporariorum* nymphs (20.5 days) compared to *B. tabaci* (19.82 days, Table 5).

It was also observed that the total reproductive days of female adult *O. strigicollis* were statistically similar with no significant difference for both whitefly species (8.29 and 7.80 days when fed *B. tabaci* and *T. vaporariorum*, respectively, Table 5). Furthermore, no significant difference was recorded in total female fecundity against both prey species; however, more eggs were laid by adult females when fed *B. tabaci* nymphs (54.18 eggs)

**Table 5  Effects of whitefly species as prey on biological traits of *O. strigicollis*.** SEs were estimated using 100,000 bootstrap. Means marked with multiple letters in the same row symbolize the significant difference using a pair bootstrap test $P < 0.05$.

| Parameters | *B. tabaci* Mean ± S.E. | *T. vaporariorum* Mean ± S.E. |
|---|---|---|
| APOP (d) | 2.35 ± 0.15 a | 2.4 ± 0.22 a |
| TPOP (d) | 19.82 ± 0.81b | 20.5 ± 0.22 a |
| Oviposition (d) | 8.29 ± 0.83 a | 7.80 ± 1.31a |
| Fecundity (F) (eggs /female) | 54.18 ± 10.52 a | 49.82 ± 15.81 a |
| Sex ratio (F: M) | 17: 9 | 11: 6 |

Notes.
APOP, Adult pre-oviposition period of female adult; TPOP, Adult pre-oviposition period of female counted from the birth.

than *T. vaporariorum* (49.82 eggs). More females were produced when fed *B. tabaci* than *T. vaporariorum*.

## Population parameters of *O. strigicollis*

The influence of the two prey species on the population parameters of *O. strigicollis* are listed in Table 6. No significant differences were found in *O. strigicollis* population parameters when fed *B. tabaci* and *T. vaporariorum* separately. The intrinsic rate of increase ($r$) and finite rate of increase ($\lambda$) for *O. strigicollis* were (0.13 d$^{-1}$ and 1.14 d$^{-1}$, respectively) when fed nymphs of *B. tabaci* and (0.11 d$^{-1}$ and 1.12 d$^{-1}$) when fed *T. vaporariorum*. However, the highest net reproductive rate ($R_0$) occurred when *O. strigicollis* was fed *B. tabaci* (30.70 offspring per female) than when fed *T. vaporariorum* (18.27). Furthermore, the mean generation time of individuals ($T$) and the values of *GRR* were higher when *B. tabaci* was offered as prey (25.46 d and 54.62) compared to *T. vaporariorum* (26.61 d and 61.57; Table 6).

## Age-stage and age-specific survival of *O. strigicollis*

The Fig. 2 explains the age-stage specific survival rate ($S_{xj}$; the possibility of a newly hatched individual that will successfully survive to age $x$ and stage $j$) of *O. strigicollis* when fed *B. tabaci* and *T. vaporariorum*. Overlap occurs between stages as a result of variations in the developmental rate of individuals. When fed nymphs of *B. tabaci,* 86.66% of *O. strigicollis* eggs successfully survived and reached to the adult stage. However, the survival rate was significantly lower (56.7%) when *O. strigicollis* were fed *T. vaporariorum* nymphs.

Age-specific survival rate ($l_x$; a simplified form of $S_{xj}$), age-stage specific fecundity ($f_x$), age-specific total fecundity of the whole population ($m_x$), and age-specific maternity ($l_x m_x$; formed on the basis of $f_x$ and $m_x$) of *O. strigicollis* when fed *B. tabaci* and *T. vaporariorum* are presented in Fig. 3. As age increased, the $l_x$ of *O. strigicollis* decreased and showed an inverse relationship for both prey species. The peak of the $m_x$ curve was at 26.98 days (5.14 eggs) and 31.09 days (5.72 eggs) when *O. strigicollis* fed on *B. tabaci* and *T. vaporariorum,* respectively. The peak of $f_x$ was at 24.90 days (8.45 eggs) and 30.88 days (7.80 eggs) for *B. tabaci* and *T. vaporariorum* nymphs respectively.

**Table 6  Population parameters of O. strigicollis.** The values of r, λ and $R_0$ were maximum when predatory bug fed on *B. tabaci* nymphs. However, *T* and *GRR* increased when *O. strigicollis* fed on *T. vaporariorum*. SEs were estimated using 100,000 bootstraps. Means marked with different letters in the same row symbolize the significant difference using a pair bootstrap test. *P* < 0.05.

| Parameters | B. tabaci (Mean ± S.E.) | T. vaporariorum (Mean ± S.E.) |
|---|---|---|
| Intrinsic rate of increase ($r$) (d$^{-1}$) | 0.1345 ± 0.0093 a | 0.1092 ± 0.0147 a |
| Rate of increase ($\lambda$) (d$^{-1}$) | 1.1439 ± 0.0106 a | 1.1153 ± 0.0162 a |
| Net reproductive rate ($R_0$) (offspring/individual) | 30.70 ± 7.5818 a | 18.2667 ± 7.0703 a |
| Generation time ($T$) (days) | 25.464 ± 0.428 a | 26.611 ± 0.753 a |
| Gross reproduction rate ($GRR$) (Offspring) | 54.62 ± 12.777 a | 61.57 ± 20.215 a |

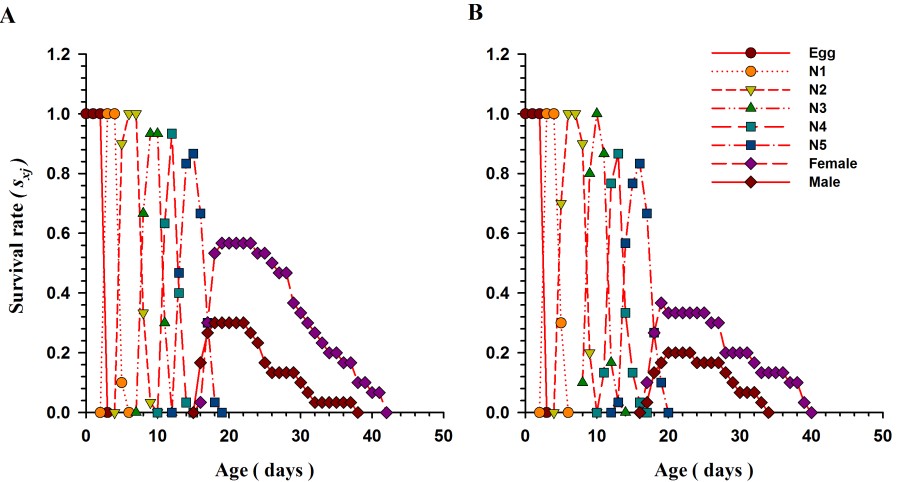

**Figure 2  Age-stage specific survival rate ($S_{xj}$) of O. strigicollis when offered with whitefly species.** The overlap occurs between stages as a result of variations in the developmental rate of individuals. When fed nymphs of *B. tabaci* (A), more eggs of *O. strigicollis* successfully survived and reached to the adult stage. However, the survival rate was significantly lower when *O. strigicollis* were fed *T. vaporariorum* (B) nymphs.

### *Life expectancy and reproductive values*

The curves for life expectancy ($e_{xj}$) of *O. strigicollis* at each stage when presented with *B. tabaci* and *T. vaporariorum* are shown in Fig. 4. It describes the estimated survival of an individual of age $x$ and stage $j$ at a later age $x$. The life expectancy of a newly hatched egg of *O. strigicollis* was 28.57 and 24.91 d on *B. tabaci* and *T. vaporariorum* respectively. The age-stage reproductive values ($v_{xj}$) of adult female *O. strigicollis* when exposed to different species of whitefly are shown in Fig. 5. Age-stage specific reproductive value is a measure of the contribution of an individual (from age $x$ and stage $j$) to a future population. When adult female *O. strigicollis* fed on *B. tabaci*, the highest peak was observed at 21 days, which was greater than when offered *T. vaporariorum* (19.79 days). The Age-stage specific reproductive curve ($V_{xj}$) indicated that the presence of *B. tabaci* had a more positive effect on *O. strigicollis* reproduction than *T. vaporariorum*.
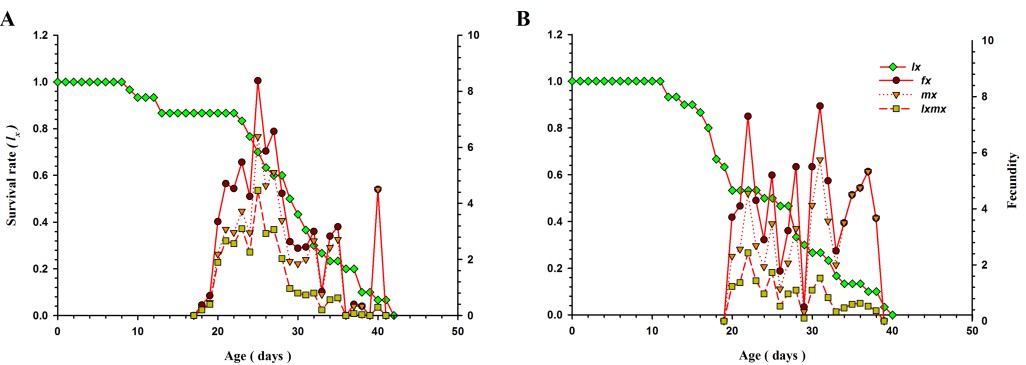

**Figure 3** **The $l_x$, $f_x$, $m_x$, and $l_x m_x$ of *O. strigicollis* presented with *B. tabaci* (A) & *T. vaporariorum* (B).** As age increased, the $l_x$ of *O. strigicollis* decreased and showed an inverse relationship for both prey species. The peak of the *mx* and *fx* curve obtained when *O. strigicollis* fed on *B. tabaci*. $l_x$: Age-specific survival rate $f_x$: female age-specific fecundity $m_x$: age-specific fecundity of the total population $l_x m_x$: and age-specific maternity.

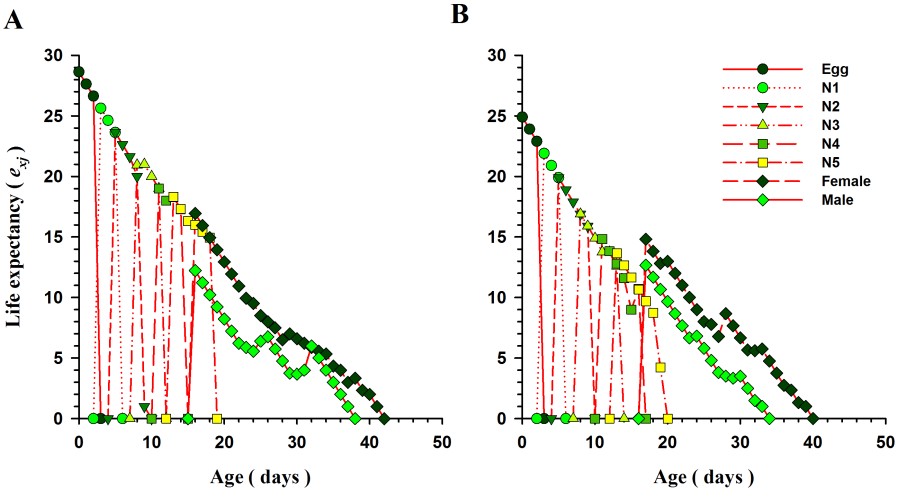

**Figure 4** **Age-specific life expectancy ($e_{xj}$) of *O. strigicollis* offered with whiteflies species.** The life expectancy of a newborn *O. strigicollis* egg was greater when fed nymphs of *B. tabaci* (A) than those of *T. vaporariorum* (B).

## DISCUSSION

To quantify the ability of a predator to combat agricultural pests, the Holling functional response model has been used for several years (*Ganjisaffar & Perring, 2015*; *Yazdani & Keller, 2016*). Handling time ($T_h$) and attack rate (*a*) are considered key parameters in explaining oscillations in predator and prey interactions (*Wang et al., 2019*). A predator's functional response to its prey plays a significant role in the effect it has on a prey population (*Begon, Harper & Colin, 1986*). Similarly, life table studies enable us to understand the ecology of an organism and supply some crucial tools to study vital biological functions such as growth, survival, and reproductive rate when an organism

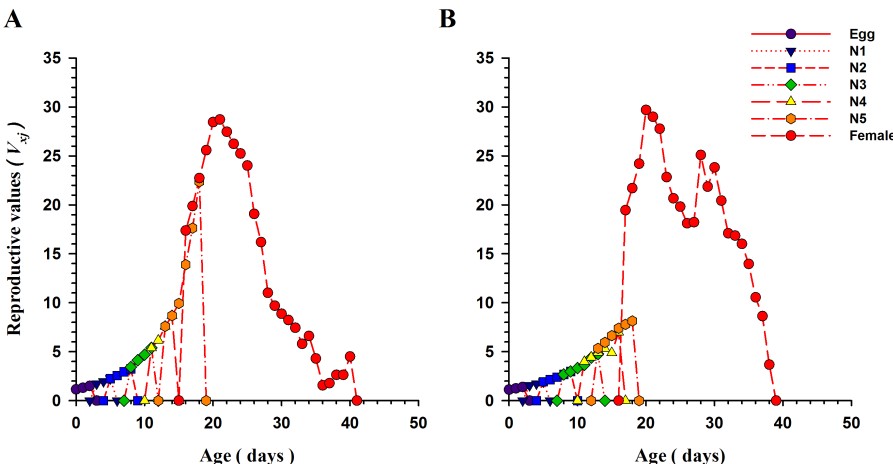

**Figure 5** **Age-stage specific reproductive values ($v_{xj}$) of *O. strigicollis* praying on whitefly species.** The reproductive curve ($v_{xj}$) indicated that the presence of *B. tabaci* (A) had a more positive effect on *O. strigicollis* reproduction than *T. vaporariorum* (B).

is in a diverse environment. Some drawbacks have been found in Jackknife methods; therefore, a bootstrap method using 100,000 resamples was developed to calculate population parameters with more accurate results (*Huang & Chi, 2012*; *Huang & Chi, 2013*). Numerous studies have investigated *Orius* spp. as a predator of *B. tabaci*, (*Arnó, Roig & Riudavets, 2008*; *Adly, 2016*; *Banihashemi et al., 2017*; *Zandi-Sohani et al., 2018*; (*Shahpouri, Yarahmadi & Sohani, 2019*).

Our results indicate that the third instar nymphs and adult female of *O. strigicollis* show type II functional responses when fed separately on six different densities of *B. tabaci* and *T. vaporariorum* third instar nymphs. With increases in prey density, the net prey consumption of both life stages of *O. strigicollis* increased until a plateau was reached. Predators with type II and III functional responses have a probability of being a stabilizing force in biological control programs (*Fernández-arhex & Corley, 2003*). *Orius* spp. have shown a type II functional response in numerous other studies (*Holling, 1965*). A type II functional response was reported for *O. albidipennis* when it was fed *T. tabaci* (*Madadi et al., 2007*), *Megalurothrips sjostedti* (Thysanoptera: Thripidae) larvae (*Gitonga et al., 2002*), and *Tetranychus turkestani* (Acari: Tetranychidae) (*Hasanzadeh et al., 2015*). In contrast to our results, the adult female *O. albidipennis* showed a type III functional response when fed *B. tabaci* third instar nymphs (*Shahpouri, Yarahmadi & Sohani, 2019*). Similarly, *M. caliginosus* showed a type III functional response when presented with nymphs of *T. vaporariorum* (*Enkegaard, Brødsgaard & Hansen, 2001*). These contradictory results may be related to changes in predator species and differences in body size. Supporting our results, *O. majusculus* and *O. laevigatus* exhibited a type II functional response when fed nymphs of *T. vaporariorum* (*Montserrat, Albajes & Castañé, 2000*). Predator functional response is influenced by several factors, such as size and density of predators and preys (*Aljetlawi, Sparrevik & Leonardsson, 2004*), temperature (*Gitonga et al., 2002*; *Zamani et al., 2006*), occurrence of alternative prey (*Abrams, 1990*), and internal state of the predator

(*Hassell, Lawton & Beddington, 1976*). In our study, the arena consisted of small Petri dishes. This small experimental arena accelerated the searching efficiency of the predatory bugs and enabled them to repeatedly attack prey that initially escaped (*Wiedenmann & O'Neil, 1991*). The optimal foraging theory of predator–prey relationships has helped to reveal the influence of different prey densities on predator handling time, searching time, and predation rate (*Cook & Cockrell, 1978*; *Stephens & Krebs, 1986*). In our study, searching time decreased with increases in prey density for both third instar nymphs and adult female of *O. strigicollis*.

To estimate the effectiveness of a predator in relation to its prey, handling time is thought to be a key parameter because it shows how long a predator takes to capture, subdue, kill, and digest a single prey item (*Atlıhan et al., 2010*). In our study, the handling time was shortest when adult female *O. strigicollis* were offered *T. vaporariorum* nymphs. In contrast to our study, the handling time was higher when adult female *O. albidipennis* fed on nymphs of *B. tabaci* (*Shahpouri, Yarahmadi & Sohani, 2019*). However, long handling time enables increased nutrient consumption from prey and hence increases the persistence of predators (*Montserrat, Albajes & Castañé, 2000*). Recent studies have shown that handling time is higher when *O. albidipennis* preys on *B. tabaci* nymphs than eggs (*Shahpouri, Yarahmadi & Sohani, 2019*). Similarly, handling time of *O. laevigatus* was longer than that for other *Orius* species when tested against different densities of thrips (*Montserrat, Albajes & Castañé, 2000*). Maximum prey consumption enhances the possibility of gaining optimal ratios between predators and pests. Hence it can be useful to accelerate the application of inoculative releases (*Wang et al., 2019*). Our results show that maximum predation occurred when adult female *O. strigicollis* fed on *T. vaporariorum*. Meanwhile, more nymphs were eaten by third instar nymphs of *O. strigicollis* when fed on *T. vaporariorum*. Higher prey densities (and thus greater prey availability) or decreases in searching area accelerate predator attack rates while reducing handling time (*Hassell, Lawton & Beddington, 1976*). In a previous study (*Opit, Roitberg & Gillespie, 1997*), lower prey densities induced reduction in predator searching activity to reduce the use of energy. In our study, no significant differences in attack rate were observed when both life stages of *O. strigicollis* preyed on nymphs of *B. tabaci* and *T. vaporariorum* separately, even with six different densities. The attack rate did not therefore depend on prey densities (*Holling, 1959*). However, our results showed higher attack rates with *B. tabaci* than *T. vaporariorum* for both third instar nymphs and adult females of *O. strigicollis*. In contrast to our results, previous work resulted in different values for handling time and attack rate when *O. majuscules* and *O. laevigatus* were fed *T. vaporariorum* (*Montserrat, Albajes & Castañé, 2000*). This may be due to changes in experimental design, environment, or predator species (*Van Alphen & Jervis, 1996*). The polyphagous predatory species of *Orius* prey on a broad range of arthropods. Additionally, prey type can significantly alter the activity of their predators (*Bonte et al., 2015*). The developmental and reproductive performance and fitness of predators in relation to particular prey types highlights their potential as active biological control agents (*Grenier & De Clercq, 2003*).

The results obtained from the age-stage two-sex life table analysis indicate that *O. strigicollis* can survive and build strong populations when feeding on different species

of whiteflies (*B. tabaci* and *T. vaporariorum*). However, we found that the nymphal developmental durations of the predator were significantly longer when praying on nymphs of *T. vaporariorum* than on nymphs of *B. tabaci*. Moreover, adult male and female longevity was longer when presented *B. tabaci* relative to *T. vaporariorum*, but a significant difference was not observed. Our results agree with previous reports indicating that the developmental period of *O. strigicollis* pre-adults is greater when they prey on *A. cracivora* than on *C. cephalonica* eggs (*Amer, Fu & Niu, 2018*). The same interaction was documented in numerous studies where the prey species strongly alter the developmental duration of pre-adults of genus *Orius* (*Kim, 1997*; *Kim, 1999*; *Ohta, 2001*; *Sengonca, Ahmadi & Blaeser, 2008*). Consequently, these fluctuations in the developmental period show that a prey species can strongly influence the developmental time of the pre-adult stages of *O. strigicollis* (*Sengonca, Ahmadi & Blaeser, 2008*). The survival of *O. strigicollis* individuals was higher when fed on *B. tabaci* nymphs than *T. vaporariorum*. *Amer, Fu & Niu (2018)* documented that higher mortality rates occurred in the pre-adult stages of *O. similis* when they fed on *C. cephalonica* eggs compared to *A. cracivora*. Similarly, the survival rate of *O. laevigatus*, *O. niger*, and *O. majusculus* individuals during development was very low (i.e., high mortality rates) when fed eggs of *E. kuehniella* than those of *F. occidentalis* (*Kiman & Yeargan, 1985*; *Tommasini, Lenteren & Burgio, 2004*). Based on our results and those of *Arnó, Roig & Riudavets (2008)*, the total developmental duration and survival of *O. strigicollis* nymphs, when fed *B. tabaci* nymphs, were similar to those of *O. majusculus* and *O. laevigatus*. *Arnó, Roig & Riudavets (2008)* also reported results similar to those reported by *Riudavets & Castañé (1998)*, suggesting that whitefly species could be considered suitable prey analogous to *F. occidentalis* larvae predated upon by *Orius* spp. Furthermore, our results showed no significant difference in adult-pre-oviposition period for both whitefly species. In contrast to our study, *Zhang et al. (2012)* found that after mating, the development of the reproductive system of adult female *O. similis* took longer (5–6 days) compared to our results. Similarly, the number of eggs laid by adult female *O. strigicollis* were comparatively higher for both whitefly species than that of *Tetranychus cinnabarinus* (*Zhang et al., 2012*).

*Chen et al. (2017)* found that intrinsic rate of increase ($r$) is a key population parameter in determining the development, growth, and survival of an organism. *Southwood & Henderson (2009)* also documented that greater values of ($r$) i.e., $r > 0$ highlight the fit of a prey with its host. Comparing our result with these studies shows that the intrinsic rate of increase was more than (0) and similar for both prey species. The net reproductive rate ($R_0$) is also considered an important demographic life table parameter. Values of $R_0$ of more than 1 indicate an increase in the mean population of an insect (*Southwood & Henderson, 2009*; *Chen et al., 2017*). Our results agree with this theory, as the highest $R_0$ was when *O. strigicollis* was fed *B. tabaci* relative to *T. vaporariorum*. In contrast to our study, low reproductive rates have been observed when *O. strigicollis* is fed *T. cinnabarinus* (*Zhang et al., 2012*). The gross reproduction rate (*GRR*) is thought to be a symbol of a rapid increase of population, which is directly related to adult eclosion and the number of eggs laid and hatched. All of these parameters can be significantly influenced by prey species (*Cocuzza et al., 1997*; *Huang & Chi, 2013*). In our results, the highest *GRR* and greatest generation

time ($T$) occurred when *O. strigicollis* fed on *T. vaporariorum* when compared to *B. tabaci*. However, in their study, *Zhang et al. (2012)* observed longer generation times at three constant temperatures when individuals of *O. similis* preyed on *T. cinnabarinus*.

The survival of a predator from the neonate to the adult stage when presented specific prey species highlights its role as an effective biological control agent (*Van Lenteren & Woets, 1988*). Similarly, the searching ability and concomitance of predators and prey in space and time, are thought to be a crucial factor in successful biological control of a pest (*Arnó, Roig & Riudavets, 2008*). *Grasselly et al. (1991)* and *Montserrat (2001)* both reported large populations of *Orius* spp. on vegetable crops where thrips and whiteflies coexist. Thus, it can be assumed that if *O. strigicollis* remains on the crop after suppression of the population of their desired prey, they can play a vital role in suppression of both whitefly species and could serve as an effective biological control agent. Supporting our hypothesis, *Riudavets (2001)* reported more than 20 *Orius* per cucumber plant when the population of *F. occidentalis* was very low, which may have been due to the presence of other pests.

In summary, our results suggest that the predatory bug *O. strigicollis* has the potential to actively maintain strong populations when in the presence of species of whitefly such as *B. tabaci* and *T. vaporariorum* and could serve as a biological control agent in cotton fields and other vegetable crops, as well as in greenhouses where the populations of these species are destructive pests. The results obtained from laboratory experiments should be useful in understanding the biology of *O. strigicollis* when in association with whitefly species, but field experiments will be required to validate them.

## ACKNOWLEDGEMENTS

The authors want to thank Miss. Abida Butt (Institute of Zoology, Punjab University Lahore, Pakistan), Mirza Abid Mahmood (College of Plant Science and Technology, Huazhong Agricultural University, Wuhan) and Fawad Zafar Ahmad Khan (Department of Entomology, University of Georgia, USA) for technical assistance.

### Funding

This work was supported by the National Natural Science Foundation of China (No. 31872023), and The National Key R&D Program of China (2017YFD0201000). The funders had no role in study design, data collection and analysis, decision to publish, or preparation of the manuscript.

### Grant Disclosures

The following grant information was disclosed by the authors:
The National Natural Science Foundation of China: 31872023.
The National Key R&D Program of China: 2017YFD0201000.

### Competing Interests

The authors declare there are no competing interests.

## Author Contributions

- Shakeel Ur Rehman and Shahzaib Ali conceived and designed the experiments, performed the experiments, analyzed the data, prepared figures and/or tables, authored or reviewed drafts of the paper, and approved the final draft.
- Xingmiao Zhou conceived and designed the experiments, analyzed the data, authored or reviewed drafts of the paper, and approved the final draft.
- Muhammad Asim Rasheed performed the experiments, analyzed the data, authored or reviewed drafts of the paper, and approved the final draft.
- Yasir Islam, Muhammad Hafeez, Muhammad Aamir Sohail and Haris Khurram analyzed the data, authored or reviewed drafts of the paper, and approved the final draft.

## Data Availability

The raw measurements are available as Supplemental Files.

## Supplemental Information

Supplemental information for this article can be found online at http://dx.doi.org/10.7717/peerj.9540#supplemental-information.

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
