# Peer review of "Predatory functional response and fitness parameters of Orius strigicollis Poppius when fed Bemisia tabaci and Trialeurodes vaporariorum as determined by age-stage, two-sex life table"

_PeerJ, doi:10.7717/peerj.9540_

## Round 0.1 · original submission · Major Revisions

Thank you for your submission. The reviewers and I believe that the manuscript is publishable in PeerJ but requires substantial revision. Please consider the reviewers' comments carefully and respond in full.

·

Basic reporting

comments inside in the document

Experimental design

comments inside in the document

Validity of the findings

comments inside in the document

Additional comments

This manuscript dealt with an interesting research topic, which the potential of using biological control
agents in the management of Bemisia tabaci and Trialeurodes vaporariorum with predator Orius similis. This will be of great interest for the readers of PeerJ journal. However, there are some concerns that need to be addressed before it gets the acceptance level.

·

Basic reporting

Included in the attached document.

Experimental design

Included in the attached document.

Validity of the findings

Included in the attached document.

Additional comments

Included in the attached document.

---

## Round 0.2 · Minor Revisions

Your revisions have been examined by two reviewers and myself. We agree that the paper is worth publishing but it requires additional clarification. Please address these minor concerns and re-submit.

·

Basic reporting

The authors answered the suggestions; the manuscript has improved a lot. congratulations; minor changes are inserted in the document

Experimental design

The authors answered the suggestions; the manuscript has improved a lot. congratulations

Validity of the findings

The authors answered the suggestions; the manuscript has improved a lot. congratulations

Additional comments

Dear author

Congratulations; the manuscript is of high quality; minor changes are inserted in the document (word)

·

Basic reporting

The paper has a clear aim, and part of comments suggested on the first review were incorporated in the new version. The proposal manuscript has merit to be published, however, some questions still arising through methodology and results that should be clarified. All questions were described in details in the following sections.

Experimental design

i) In the first submission I asked about: i) how the authors did the model selection?, ii) How they check model assumptions?, and iii) It will be important at least include a half-normal plot of the residuals. The answer addressed to my question was:
"Dear sir as we used polynomial logistic regression for proportion of prey killed/ consumed so the errors associated with such as variable are likely to be distributed binomially. "

I disagree with the authors, and I am still concerned about goodness-of-fit of their model. Proportion data may present overdispersion, i.e., greater variability than expected by the binomial models. A possible solution to check the goodness-of-fit and verify the assumptions of this model can be addressed through half-normal plots with a simulated envelope. Please, included the half-normal plot as supplementary material to certify that the model is well fitted to the data. See reference bellow:

Half-Normal Plots and Overdispersed Models in R : The hnp Package. DOI: 10.18637/jss.v081.i10.

ii) The authors reported that who attached, as supplementary material, a plot of residuals versus fitted values, a Q-Q plot of empirical cumulative distribution function of a data set and a specified theoretical cumulative distribution function (probably normal distribution). However, these documents were not attached in the supplementary material. Please, supply them.

Validity of the findings

i) Table 1 shows there are no reasons to include more than an intercept parameter to explain T. vaporariorum. Hence, how the author could explain the inclusion of linear, quadratic, and cubic effects? It should be proper remove those effects as p-value is much higher than the confidence level adopted. Furthermore, they should update results and discussion after model selection.

ii) Why the Adult attack rate was zero when fed with 14 prey?

iii) Lines 376-377 Authors could discuss in more details results about searching time demonstrating how their findings is related to recent research and why this variable has a practical value.

Additional comments

i) I recommend to the authors include a smaller size of points in the Figures 2, 4, and 5 in order to facilitate the visualization of them.

---

## Round 0.3 · accepted · Accept

Thank you for addressing the issues raised by reviewers. Both the reviewers and I agree this paper is ready for publication.

·

Basic reporting

See General comments

Experimental design

See General comments

Validity of the findings

see General comments

Additional comments

Authors reviewed and included all diagnostic plots, and they addressed the statistical issues as recommended. They have done a great job and I fully accept their paper.